# Antibody Therapeutics as Interfering Agents in Flow Cytometry Crossmatch for Organ Transplantation

**DOI:** 10.3390/jpm13061005

**Published:** 2023-06-16

**Authors:** Michael L. Kueht, Laxmi Priya Dongur, Muhammad A. Mujtaba, Matthew F. Cusick

**Affiliations:** 1Department of Surgery, Multiorgan Transplant and Hepatobiliary Surgery, University of Texas Medical Branch, 301 University Boulevard, Galveston, TX 77555, USA; mlkueht@utmb.edu (M.L.K.); ladongur@utmb.edu (L.P.D.); 2Department of Medicine, Transplant Nephrology, University of Texas Medical Branch, 301 University Boulevard, Galveston, TX 77555, USA; mamujtab@utmb.edu; 3Department of Pathology, Division of Histocompatibility and Immunogenetics, University of Michigan Medicine, 2800 Plymouth Rd., Building 36, Ann Arbor, MI 48109, USA

**Keywords:** flow cytometry, transplantation, personalized medicine, monoclonal antibodies, HLA, histocompatibility, cross match

## Abstract

Donor–recipient matching is a highly individualized and complex component of solid organ transplantation. Flowcytometry crossmatching (FC-XM) is an integral step in the matching process that is used to detect pre-formed deleterious anti-donor immunoglobulin. Despite high sensitivity in detecting cell-bound immunoglobulin, FC-XM is not able to determine the source or function of immunoglobulins detected. Monoclonal antibody therapeutic agents used in a clinic can interfere with the interpretation of FC-XM. We combined data from the prospectively maintained Antibody Society database and Human Protein Atlas with a comprehensive literature review of PubMed to summarize known FC-XM-interfering antibody therapeutics and identify potential interferers. We identified eight unique FC-XM-interfering antibody therapeutics. Rituximab (anti-CD20) was the most-cited agent. Daratumuab (anti-CD38) was the newest reported agent. We identified 43 unreported antibody therapeutics that may interfere with FC-XM. As antibody therapeutic agents become more common, identifying and mitigating FC-XM interference will likely become an increased focus for transplant centers.

## 1. Introduction

In solid organ transplantation, donor–recipient matching is of paramount importance. This compatibility must occur on several levels, including ABO- and human leukocyte antigen (HLA)-typing. Histocompatibility testing guides the assessment and risk-stratification of allo-immune responses. The degree of matching between donor and recipient human leukocyte antigens (HLAs) is determined, and the transplant candidate is tested to identify whether antibodies against the mismatched donor antigens are present [1,2,3,4,5]. Since the discovery in the 1960s of the major role of pre-formed anti-HLA antibodies in donor-directed cytotoxicity and post-transplant outcomes, several methodologies have emerged to detect these deleterious antibodies [6]. 

The fact that the major histocompatibility complex (MHC) genes that code for HLA proteins are some of the most variable regions of the genome, coupled with the severe consequences of acute allo-rejection, have pushed testing for the presence and function of anti-HLA antibodies to become quite sophisticated [7,8]. Over the past 40 years, the standard in donor–recipient HLA compatibility testing has progressed from the complement-dependent cytotoxicity crossmatch (leukoagglutination and donor lymphocyte death when incubated with recipient serum) to a combination of high-resolution antibody detection with solid-phase antigen-coated bead technology and flow cytometry crossmatching (FC-XM) [9].

First reported in 1983 [10], FC-XM is the most sensitive detection assay. FCXM can detect antibodies independently from complement-fixation and cell death. The assay is performed by incubating patient sera with purified donor lymphocytes and detecting the antigen–antibody complex using a secondary fluorochrome-conjugated anti-human IgG monoclonal antibody. The strength of the reaction is given as the mean channel shift (MCS) from the fluorescence of the same donor cells when incubated with a known HLA-negative human serum. FC-XM is a semi-quantitative assay that can simultaneously detect the presence of HLA antibodies binding to both T- and B-cells.

As with the complement-dependent cytotoxicity XM, FC-XM relies on mixing donor lymphocytes and recipient serum, but with the addition of fluorescent-labeled anti-IgG antibodies (anti-human Fc-fluorochrome anti-serum). Additionally, FC-XM does not rely on the assessment of cytotoxicity, only the excitation of immunoglobulin-bound fluorescent molecules. One of the main advantages attributed to FC-XM is the ability to produce a semi-quantifiable result with fluorescent intensity values as opposed to a subjective evaluation of leuko-agglutination and cell viability, as in the complement-dependent cytotoxicity XM [11]. In the strength of FC-XM also lies its weakness; despite its high sensitivity in detecting cell-bound immunoglobulin, FC-XM is not able to determine the source or functional status of the immunoglobulins detected [12,13].

Monoclonal antibody therapeutic agents used in a clinic for various indications can interfere with the interpretation of antibody binding and identification in FC-XM [14]. These agents tend to bind lymphocyte cell surface proteins, providing an interfering Fc domain to which the fluorescent-labeled anti-IgG may bind. Because a ‘positive’ FC-XM result due to anti-HLA antibodies is associated with a poor transplant outcome, the interference of monoclonal antibody treatments causing a false positive FC-XM may unnecessarily deny a patient the opportunity to receive a transplant.

Biologic agents used in the desensitization of transplant candidates with high levels of anti-HLA antibodies (rituximab, anti-thymocyte immunoglobulin, IVIG, and alemtuzumab) were the first to be identified as creating false positive results with FC-XM and some strategies have been employed with varying success to overcome this effect [15,16]. As biologic agents to treat a host of diseases become more prevalent in a clinic, the issue of biologically influenced false positive FC-XM results will also become more common [13].

Disqualification from transplantation due to a false positive result leads to a longer time on a waitlist, which can only lead to poor clinical outcomes and also to an increased economic burden on the healthcare system. Some estimates show that twenty percent of patients undergoing crossmatch have false positive results [14]. Clinicians may be inclined to avoid HLA-incompatible transplantation due to a potential increase in morbidity and mortality as well as increased costs compared to HLA-compatible transplantation [15]. However, a longer time spent on dialysis continues to be a healthcare burden far superior to transplantation. Therefore, understanding the molecular mechanism behind the unacceptably high false positive rates can potentially mean less of a cost of care, shorter times on waitlists, less morbidity, better quality of life, and better mortality. 

Here, we summarize the known and potential FC-XM-interfering antibody therapeutic agents and describe the potential FC-XM interference of daratumumab, a monoclonal antibody used in the treatment of multiple myeloma.

## 2. Methods

### 2.1. Known Biologic Interferers

In order to document the published literature on antibody therapeutics causing FC-XM interference, a systematic literature search was undertaken in the National Library of Medicine via PubMed for peer reviewed published reports. The search terms used included combinations of the following: “biologic”, “monoclonal antibody”, “interference”, “false positive”, “flow cytometry”, “crossmatch”, and “transplant”. Generic pharmaceutical names, brand names, known or proposed targets, and potential countermeasures taken against the interference were noted if documented. Additionally, the subtype of immunoglobulin and any post-translational modifications were recorded. Published reports were excluded if FC-XM was not explicitly mentioned or no detailed information was provided on the FC-XM process. Reports were only included if there was a specific mention of interference in FC-XM.

### 2.2. Potential Biologic Interferers

Because those antibodies most likely to potentially interfere with FC-XM are those that interact with T- and B-cells, a search of European-Union- and United-States-approved antibody therapeutics was undertaken in the Antibody Society database [17]. This prospectively maintained comprehensive list is comprised of approved antibody therapeutics and those in regulatory review in the European Union (EU) or United States (US). Included in the list are products that were granted approval but subsequently withdrawn from the market. Lymphocyte (T- and B-cell) cytoplasmic and cell surface expression were documented via a crossreferencing search in the Human Protein Atlas [18]. This prospectively maintained open access online resource based in Sweden documents the distribution of all known proteins discovered by the Human Genome Project across all major tissues and organs in the human body, including immune cells. Where known, the protein location on or in the cell was documented. 

## 3. Results

### 3.1. Known Biologic Interferers

Eight unique FC-XM-interfering antibody therapeutics were identified via PubMed in ten published reports. All known and documented FC-XM interferers were agents that had been used to desensitize transplant candidates with existing anti-HLA antibodies. Humanized and chimeric monoclonal IgGs were the most common formulation with six of the eight agents. Two agents were polyclonal IgG products. Intravenous IgG (IVIG), an agent consisting of pooled purified human IgG, is unique in that there is no specific target for the included immunoglobulins. These specificities are based on exposures of the human donors and have been shown to change with time [19]. Thymoglobulin is a polyclonal IgG created by inoculating rabbits with homogenates of human pediatric (immature) thymus, thus creating an agent with a wide range of targets on thymocytes and lymphocytes. Rituximab (anti-CD-20) was the most-cited interfering agent with the first report documented in 2001. This agent targets a protein found on the surface of mature B-cell lymphocytes and is used in blunting undesirable intrinsic antibody responses such as those in organ transplantation. Infliximab (anti-TNFa), commonly used in Crohn’s disease, was tested in one report and was found to not interfere significantly with FC-XM [16]. Known FC-XM-interfering biologic agents are summarized in Table 1 [20,21,22,23,24,25,26,27,28,29].

### 3.2. Proposed Mechanisms of Interference

No mechanisms of FC-XM interference were definitively proven in our literature search, leaving only theoretical and logical assumptive mechanisms. These proposed mechanisms are partly based on the chemical nature of successful countermeasures and the working principles of flow cytometry. The most commonly proposed potential mechanism of FC-XM interference was noted to be the provision of additional cell-bound immunoglobulin Fc regions, resulting in additional epitope binding sites for the anti-human Fc-fluorochrome anti-serum [20,21,23]. The components of an antibody include two light chains and two heavy chains, whereby each of which have a variable (Fab) and a constant (Fc) portion. The variable portions of the heavy and light chains form an antibody-binding region, also known as the epitope. FC-XM interference is mediated primarily by these primary portions [32]. For instance, the Fab fragment of Rituximab binds to the CD-20 receptors located on B-cells. This fixation leads to downstream immunologic effects causing B-cell lysis, increasing the soluble antibodies, as opposed to binding to the cells [20]. This resultant excess of fluorescent proteins in the cell suspension results in more activation in the sheath and fluorescent emission captured, leading to an increase in channel shifts and an interpretation of the result as “positive”. Additional mechanisms proposed in the literature involve direct HLA antigen binding with IVG and complement activation, leading to indirect IgG deposition [16,20]. Regardless of the upstream initiating signal, the resultant excess of non-pathologic IgG detected via FC-XM appears to be the functional description of FC-XM interference. 

### 3.3. Proposed Countermeasures of Interference

Interventions to counteract FC-XM interference include non-specific chemical denaturing and enzymatic agents (dithiothreitol (DTT), pronase) and highly specific anti-idiotype blocking antibodies (anti-rituximab, antialemtuzumab). The treatment of candidate sera with DTT, a thiol redox reagent, was documented to change the FC-XM result from “positive” to “negative” in the presence of rituximab, eculizumab, intravenous IgG (IVIG), anti-thymocyte globulin (ATG), and daratumumab [21,33]. The proposed mechanism of the DTT treatment of sera is based on its ability to interrupt proteins’ tertiary structures by reducing the disulfide links. This function of DTT is employed in denaturing IgM (commonly present as pentamer or hexamer conglomerates), but usually leaves IgG intact, depending on the concentration and duration of interaction [34]. Other than the fact that protein structures differ in the amount of accessible disulfide bonds, it was not proven why DTT treatment would neutralize IgG antibody therapeutics such as rituximab [20,21]. However, the analysis of FC-XM cannot provide information regarding other mechanisms of counter-interference, given that rabbit Ab (ATG) is not detected by the assay [20]. DTT may interfere with aspects of the antibody-mediated complement activation that is not detected via standard FC-XM assays, potentially improving the false positive rates. Drawbacks to DTT treatment include the non-specific nature of the protein denaturation and the high doses used, potentially leading to false negative FC-XM results [30]; see Figure 1.

The addition of pronase, a mixture of non-specific bacterial proteases, to donor lymphocyte preparations was documented to reduce FC-XM interference in the presence of rituximab [21]. The proposed mechanisms for the pronase treatment of cells included the cleavage of B-cell-bound Fc receptors, which would participate in binding to the antibody therapeutic interferer. Cell surface Fc sites as the true targets of pronase in FC-XM have been called into question. Drawbacks to pronase treatment include the non-specific nature of protein degradation and alterations in cell surface HLA pattern expression, also potentially leading to false negative results [30,31].

The treatment of candidate sera with specific anti-idiotype antibodies against rituximab and alemtuzumab was reported to successfully counteract FC-XM interference [21]. The proposed mechanism of anti-idiotype antibody treatment was the binding of the interfering IgG in solution, preventing it from binding to the donor cells during incubation for FC-XM. This effect was reported to be very specific and was shown to not cause “false negative” FC-XM results [21]. The feasibility and cost of such methods were not documented. The methods used to overcome FC-XM interference are summarized in Table 1. 

### 3.4. Potential Biologics

As of January 2022, 136 antibody therapeutics were approved or pending regulatory approval in the United States (US) and the European Union (EU). The first antibody therapeutic to be approved anywhere in the world for humans was Muromonab-CD3, also known as OKT3. This medication was indicated for the treatment of acute rejection in organ transplantation and was discontinued in 2010 due to the availability of superior options [35]. From 1998 to 2012, yearly antibody therapeutic approvals were relatively stable with an exponential annual increase between 2012 and 2022. Through crossreferencing the Antibody Society and Human Protein Atlas databases, fifty-one unique antibody drugs with T- and/or B-cell target protein expression were identified. The location within the cell, membrane-bound vs. intra-cellular, was not well known for all drug targets. This list included the eight known biologic FC-XM interferers. Therefore, an additional 43 hitherto unreported antibody therapeutics that may contribute to false positive FC-XM were identified (Table 1). Cancer was the most common disease process indication (58%), followed by autoimmune diseases (38%). Interestingly, transplant was rarely an approved indication, often being used off-label from the FDA-approved indications.

### 3.5. Rituximab Interference

Rituximab is a chimeric anti-CD20 IgG1 monoclonal antibody approved for the treatment of non-Hodgkin lymphoma. It has been used extensively in transplantation as a desensitizing agent since 2002, including for ABO-incompatible transplantation and for the treatment of auto-immune kidney disease since 2009. The CD20 protein is a membrane-bound protein present in the highest concentration on non-terminally differentiated B-cell lymphocytes; plasma cells express CD20 to a much lesser extent. The interference of rituximab in the FC-MX is well documented, but the mechanism has not been fully elucidated, similarly to the ambiguity that persists as to the exact mechanism of CD20 signaling. Because rituximab is widely used in diseases that may lead to the need for kidney transplantation, encountering this agent as an interferer in FC-XM has been the hitherto most commonly published description. 

### 3.6. Obinutuzumab Interference

Obinutuzumab is a glycoengineered anti-CD20 IgG1 monoclonal antibody approved for the treatment of chronic lymphocytic leukemia and follicular lymphoma. The primary mechanism of action is via the activation of antibody-dependent cellular cytotoxicity. A recent prospective randomized controlled trial evaluated the effect of obinutuzumab in direct comparison to RTX in highly sensitized end-stage renal disease patients undergoing evaluation for transplantation. Crossmatch assays were conducted at the start of the protocol and 2 weeks after. A secondary outcome was that obinutuzumab did appear to interfere with FC-XM, whereas RTX interfered considerably, resulting in false positive crossmatch results [36]. Therefore, although obinutuzumab was suspected to interfere with FC-XM due to its similar target to rituximab, other factors appeared to be influencing its ability to be detected via FC-XM. 

### 3.7. Daratumumab Interference

Daratumumab is the first human-specific anti-CD38 IgG1 monoclonal antibody approved for the treatment of multiple myeloma [37]. Daratumumab has long been known to interfere with red blood cell crossmatching for transfusion in multiple myeloma patients. Mechanisms to counteract the interference of daratumumab in transfusion medicine have focused on the DTT treatment of donor cells to induce the cleavage of cell surface antigens. This approach was opposite to the described countermeasures of the known FC-XM interferers, wherein candidate sera was the component treated with DTT [29,33,34,38,39]. Our group recently reported a case of a transplant candidate with well-controlled multiple myeloma in daratumumab therapy, with persistently positive FC-XM results against most donors. After suspecting interference and carefully employing DTT and pronase, accurate crossmatching was able to be achieved and she was transplanted successfully. The negative crossmatch, however, could have potentially been a result of not only variable CD38 expression but also an inadvertent exclusion of B-cells that were tagged with FITC anti-IgG [40,41,42]. 

Of all the known biologic interfering agents, rituximab continues to be the most detected in part due to its longer presence in the market leading to increased research in understanding all aspects of its mechanism. Other biologics, such as obinutuzumab and daratumumab, continue to have an important clinical effect, given the preliminary research showing crossmatch interference with possible solutions for counter-interference to mitigate false positive results. 

## 4. Discussion

In transplantation, the primary application of FC-XM is to identify pathologic donor-reactive sensitization of the recipient that could be a risk for allograft rejection. FC-XM has supplanted the basic complement-dependent cytotoxicity crossmatch due to increased sensitivity when assessing the presence of donor-specific anti-HLA antibodies and the increased availability of flow cytometry technology. Despite the success of FC-XM in revolutionizing donor–recipient matching, weaknesses in its methodology remain.

Normally, a patient’s current serum (within 30 days of transplant) is used in the crossmatch as well as a historical serum sample. For sensitized patients, the peak serum is the historical sample with the highest amount of anti-HLA antibodies. Each crossmatch scenario takes into consideration the potential recipient’s past sensitizing events, including pregnancies, previous transplants, and blood transfusions. The crossmatch test results are carefully reviewed in combination with the patient’s HLA antibody screening results and anti-HLA antibody specificities, which may be donor-specific.

FC-XM involves incubating a potential recipient’s serum with the potential donor’s lymphocytes, either isolated from peripheral blood or the donor spleen. The mixtures of cells and serum are incubated with a secondary antibody used to identify the human immunoglobulin that is specific to the donor cells. Labs generally use fluorescein-labeled antibodies against human IgG (anti-human IgG F (ab)/FITC), T-cell markers (e.g., CD3 fluorescent marker) and B-cell markers (e.g., CD19 fluorescent marker) [29,33].

The lymphocyte XM result provides an assessment of how much of the IgG antibody binds to the donor cells. Historically, the XM was performed to prevent hyperacute/acute rejection, which has become a mostly historical phenomenon with an increased ability to detect donor-specific anti-HLA antibodies pre-transplant. Nonetheless, the sensitivity of the assay has also increased, from the complement-dependent cytotoxicity XM to the detection of HLA antibody binding via flow cytometry (F-XM). FC-XM cut-offs are expressed as mean channel shifts (MCSs) and may show differences between labs based on the instruments and specific reagents that they use. It is important to appreciate that antibodies binding to a lymphocyte target do not necessarily represent the recognition of donor HLA antigens but recipient antibodies. The combination of physical XM, virtual XM, and in-depth antibody testing via solid phase assay allows for a much better appreciation of the nature of positive/negative responses. With the addition of pre-transplant assessment of the patient’s immune status (including potential memory), well-performed XM can allow for judicious risk stratification, which can guide patient management before and after transplantation.

Modified assays for crossmatching have been proposed to overcome the false positive results encountered during FC-XM. Complement-dependent cytolytic FC-XM (cFC-XM) combines aspects of the original CDC assay and FC-XM. It creates a dichotomy between the lytic and non-lytic antibodies that are formed when recipient serum is crossed with donor serum. This aims to allow for the understanding of donor cell viability in response to recipient antibodies, as not all antibodies lead to a lytic reaction leading to cell injury and rejection. Saw et al. demonstrated that cFC-XM is more sensitive than standard FC-XM for the detection of clinically relevant recipient antibodies against donor cells, based on the significant differences in cell viability noted via cFC XM. Thammanichanond et al. showed that transplant recipients with antibody-mediated rejection had retrospectively positive cFC-XM results, despite negative pre-transplant standard FC-XM results, showing that the evaluation of transplant candidacy and graft rejection rates may be better understood with modified FC-XM assays [16,32,43]. This technology has yet to be widely adapted, and the cost–benefit analysis remains unknown.

IgG antibody therapeutics used to treat patients for related (or unrelated) conditions can significantly influence the FC-XM result, mainly as a false positive. The rapidly expanding list of antibody therapeutics will inevitably increase the frequency with which biologics may interfere with FC-XM in donor–recipient compatibility testing in solid organ transplantation. Despite this growing concern, efforts to prevent FC-XM interference have been notably limited, focusing mainly on non-specific techniques (DTT, pronase) that lead to a host of other issues with test interpretation and may swing the pendulum from false positive to false negative.

The physical and chemical characteristics of the antibody therapeutics influence their effects on immunogenicity. Post-translational modifications may significantly influence affinity and off-target effects. While most monoclonal antibodies are glycoengineered to incorporate glycosylation on the fragment crystallized (Fc) portion of the antibody, some antibodies are not. The Fc portion is generally responsible for mediating the anti-inflammatory mechanisms. Based on the modulation of the Fc, the antibody may have a stronger affinity for antibody-dependent cell-mediated cytotoxicity instead of complement-dependent cytotoxicity or antibody-dependent cell-mediated phagocytosis. It is this variation in effector function, dictated by the glycosylation, galactosylation, or sialylation, that has downstream distinctions in the ultimate profile of the antibody. Furthermore, whether the Fc-glycan is homogenous or heterogenous impacts the antibodies’ predilections to the pathways employed for the desired cytotoxicity. As such, modulated Fc regions interfere with FC-XM assays differently, as was seen in the different interaction profiles of the previously mentioned antibodies, such as the different behavior of the anti-CD20 antibodies obinutuzumab and rituximab. Ultimately, it is imperative to understand the glycoengineered properties and other post-translational modifications of the antibodies to adequately assess the results of FC-XM in the setting of chronic or acute antibody use [32]. 

## 5. Conclusions

In this paper, we have summarized the known antibody therapeutic FC-XM interferences and reported methods to counteract the interferences. In practice, interpreting FC-XM in transplantation involves a host of Appendix A in addition to fluorescent intensity data. The presence or absence of anti-HLA antibodies as determined via solid-phase antigen-coated bead technology is a crucial factor in putting the FC-XM results into perspective, providing a signal for the need to check for biologic inference. If the FC-XM result is positive and there are no detectable anti-HLA antibodies, interference is to be highly suspected. This is especially important given that laboratories often do not know which medicines a patient is taking.

Since currently more patients are prescribed rituximab for the management of autoimmune diseases and a history of antibody exposure, this biologic continues to have bigger clinical implications in comparison to other biologics. However, with increasing prescriptions of other biologics, more transplant patients are likely to have a broad range of medications in their medical history that can lead to interference at the time of transplant evaluation.

Additionally, we have compiled a resource of potential FC-XM interferers based on the available antibody therapeutics, a list that grows daily. We suspect that only agents with targets (intended or not) on T- and B-cells will interfere significantly with FC-XM, given that the cell preparations are purified to various degrees and this assay is designed to detect anti-HLA antibodies. The interesting finding that infliximab did not interfere with FC-XM supports this hypothesis.

Given that transplant candidacy is at stake because of FC-XM results, it is important to continue various avenues of research to fully understand the nuances of interaction of monoclonal antibodies and the currently utilized assays. If the assays are developed to become more reliable within the context of the comorbidities of the patient, it will improve not only patient-centered outcomes but will also reduce the cost of care and the healthcare burden that patients with end-stage renal disease have on the system. Since antibody therapeutic FC-XM interference will likely become a more common phenomenon as monoclonal antibodies become more popular and assays are more widely used, methods to ensure accurate assay results will need to be pursued. This will allow for a personalized assessment of each individual patient to provide them with the best chance possible of securing a well-matched organ that will significantly increase their quality of life.

## Figures and Tables

**Figure 1 jpm-13-01005-f001:**
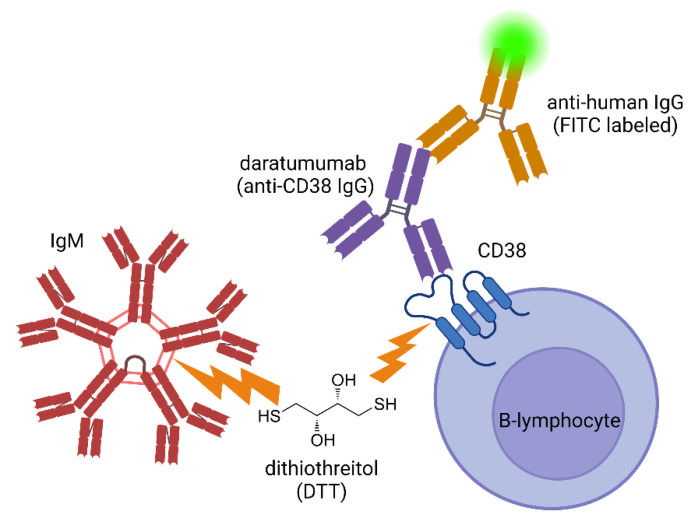
Potential mechanism of dithiothreitol (DTT) treatment as an antidote to daratumumab inference in FC-XM. DTT can reduce thiolated proteins, breaking disulfide bonds such as those found binding IgM molecules in the circulating pentamer state. Disulfide bonds are also present in the CD38 protein structure. Dissolving them may cleave the daratumumab–CD38 complex from the B-cell surface, preventing FITC-labeled anti-human IgG from remaining in the cell suspension used for flow cytometry crossmatching. Altering the cell surface protein landscape may have the drawback of creating false negative crossmatch results. Created using BioRender.com.

**Table 1 jpm-13-01005-t001:** Known FC-XM interferers. (Dtt—Dithiothreitol, anti-RTX-Ab—anti-rituximab antibody, anti-ALM ab—anti-alemtuzumab antibody).

Agent	Brand Name(s)	Target/Format	Proposed Antidote	References
Rituximab	MabThera, Rituxan	CD20; chimeric IgG1	dtt serum, pronase cells, anti-RTX-Ab	[9,10,13,14,15,16,17,20]
Daclizumab	Zinbryta; Zenapax	IL-2R; humanized IgG1		[10]
Eculizumab	Soliris	C5; humanized IgG2/4	dtt sera	[9]
Alemtuzumab	Lemtrada; MabCampath, Campath-1H	CD52; humanized IgG1	anti-ALM ab	[10,14,21]
Basiliximab	Simulect	IL-2R; chimeric IgG1		[13]
Daratumumab	Darzalex	CD38; humanized IgG1	dtt sera	[30,31]
Intravenous IgG (IVIG)	10 different brand names	Multiple targets; concentrate of pooled human IgG *	dtt sera	[9]
Anti-thymocyte globulin (ATG, rabbit)	Thymoglobulin	Multiple targets (incl CD4, CD152); polyclonal xeno-IgG *	dtt sera	[9]

* IVIG with polyclonal immunoglobulins do not have specific targets and are donor dependent.

## Data Availability

Not applicable.

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
