# Peer review of "Antibody Therapeutics as Interfering Agents in Flow Cytometry Crossmatch for Organ Transplantation"

_jpm, 2023, doi:10.3390/jpm13061005_

Round 1

Reviewer 1 Report

Authors present an interesting review where summarized known and potential FC-XM-interfering antibody therapeutic agents.

Minor concerns

1-     Clarify the purpose and importance of the information. The introduction could benefit from a brief explanation of why knowing about biologic interferers and their potential mechanisms and countermeasures is important. Please add a paragraph stating the impact of inaccurate FC-XM results on transplantation outcomes or patient safety.

2-     Provide more context for the proposed mechanisms of interference: The section on proposed mechanisms of interference could be expanded to provide more context for readers who may not be familiar with the technical terms and concepts. This could involve briefly explaining what an epitope is, what cell-bound immunoglobulin Fc regions are, and what indirect IgG deposition entails.

3-     Discuss the limitations of the studies. The information provided on the proposed countermeasures of interference could benefit from a discussion of the limitations of the studies that have been conducted. For example, the section on DTT treatment could mention that the reason why it is not clear why DTT neutralizes IgG antibody therapeutics such as rituximab is because the specific mechanism is not yet understood.

4-     Provide a summary of key findings. It may be helpful to provide a brief summary of the key findings at the end of results section, to help readers quickly understand what has been presented.

5-     Consider adding visual aids. Figures can be useful in presenting complex information in a more accessible way. The information on the proposed mechanisms of interference for example.

6-     Clarify the implications. While the text briefly mentions the implications of the findings, it could benefit from a more detailed discussion of what the findings mean for clinical practice. Specifically, it could address whether OBZ and daratumumab interference are likely to be common issues in transplant and transfusion medicine, and how clinicians should approach these issues in practice.

Minor editing of English language required

Reviewer 2 Report

In this paper, Kueht and colleagues discussed the spectrum of biological agents that affect FCXM results. It is an essential issue for the HLA laboratories, as monoclonal antibodies are commonly used in patients awaiting organ transplantation. The article is well-written and provides all the essential information. The authors discussed all of the frequently used drugs and provided proposed action and antidote that is of special importance. Please consider discussing how bio-drugs may affect FCXM with cytotoxicity assessment by viability dye. (https://pubmed.ncbi.nlm.nih.gov/23375279/ ; https://pubmed.ncbi.nlm.nih.gov/18454488/ ; https://pubmed.ncbi.nlm.nih.gov/22310580/)

Author Response

  1. Added per comments with the included references.
  2. Modified assays to FC-XM have been proposed to overcome the false positive rates. The complement dependent cytolytic FC-XM (cFC-XM) creates a dichotomy between the lytic and non-lytic antibodies that are formed when recipient serum is crossed with donor serum. This allows to understand donor cell viability in response to recipient antibodies, as not all antibodies lead to a lytic reaction leading to cell injury and rejection. Saw et al demonstrated that cFC-XM is more sensitive than standard FC-XM in detection of clinically relevant recipient antibodies against donor cells based on the significant differences in cell viability noted on cFC-XM. Thammanichanond et al showed that recipients with positive cFC-XM were more likely to have incidence of Antibody mediated rejection compared to recipients with positive standard FC-XM, showing that evaluation of transplant candidacy and graft rejection rates may be better understood with modified FC-XM assays.